# Physiological Changes in Chicken Embryos Inoculated with Drugs and Viruses Highlight the Need for More Standardization of this Animal Model

**DOI:** 10.3390/ani12091156

**Published:** 2022-04-29

**Authors:** Simone Sommerfeld, Antonio Vicente Mundim, Rogério Reis Silva, Jéssica Santos Queiroz, Maisa Paschoal Rios, Fabiana Oliveira Notário, Alessandra Aparecida Medeiros Ronchi, Marcelo Emílio Beletti, Rodrigo Rodrigues Franco, Foued Salmen Espindola, Luiz Ricardo Goulart, Belchiolina Beatriz Fonseca

**Affiliations:** 1School of Veterinary Medicine, Federal University of Uberlândia, Uberlândia 38402-018, Brazil; antoniomundim@ufu.br (A.V.M.); rogerioreissilva98@gmail.com (R.R.S.); jesk.queiroz@hotmail.com (J.S.Q.); maisapaschoal@hotmail.com (M.P.R.); fabiana.notario@hotmail.com (F.O.N.); alessandra.medeiros@ufu.br (A.A.M.R.); biafonseca@ufu.br (B.B.F.); 2Institute of Biomedical Sciences, Federal University of Uberlândia, Uberlândia 38405-319, Brazil; mebeletti@ufu.br; 3Institute of Biotechnology, Federal University of Uberlândia, Uberlândia 38405-319, Brazil; rodrigorfr@yahoo.com.br (R.R.F.); foued@ufu.br (F.S.E.); lrgoulart@ufu.br (L.R.G.)

**Keywords:** *in ovo*, chorioallantoic membrane, *Gammacoronavirus*, toxicity, oxidative stress

## Abstract

**Simple Summary:**

Over the years, the chicken embryo (CE) has been a widely used animal model, which is essential to decrease the number of born animals in experiments. Thus, we intended to know if there is a lack of standardization in using embryos in research. Therefore, the objective of this study was to verify whether alterations in CE of different ages are specific to the model, having as reference a virus and two drugs that cause known alterations in adults and other species. The response of embryos to challenges with viruses and drugs did not always occur as expected compared with adult animals. Although macroscopic and microscopic changes were visible in the infected group, other laboratory analyses did not show significant changes. Our results showed that some drugs and viruses can generate laboratory results that seem to be inherent to the model studied and depend on the CE’s developmental stage.

**Abstract:**

Several studies have been developed using the *Gallus gallus* embryo as an experimental model to study the toxicity of drugs and infections. Studies that seek to standardize the evaluated parameters are needed to better understand and identify the viability of CEs as an experimental model. Therefore, we sought to verify whether macroscopic, histopathological, blood count, metabolites and/or enzymes changes and oxidative stress in CE of different ages are specific to the model. To achieve this goal, *in ovo* assays were performed by injecting a virus (*Gammacoronavirus*) and two drugs (filgrastim and dexamethasone) that cause known changes in adult animals. Although congestion and inflammatory infiltrate were visible in the case of viral infections, the white blood cell count and inflammation biomarkers did not change. Filgrastim (FG) testing did not increase granulocytes as we expected. On the other hand, CE weight and red blood cell count were lower with dexamethasone (DX), whereas white blood cell count and biomarkers varied depended on the stage of CE development. Our work reinforces the importance of standardization and correct use of the model so that the results of infection, toxicity and pharmacokinetics are reproducible.

## 1. Introduction

The embryonated egg is self-sufficient, and its natural development at 37 °C and 60% humidity guarantees the maintenance of these animals without more complex means of support. Furthermore, within the egg, the embryo is a highly controlled, accessible and relatively transparent model in which normal physiology, disease pathology and the consequences of experimental manipulation can be easily visualized [1], allowing high reproducibility and simplified and economical experiments [2].

Numerous studies have been carried out using the *Gallus gallus* embryo as an experimental model for evaluating the toxic effects of drugs and infections. These studies usually evaluated blood, allantoic and amniotic fluid biochemical parameters, erythrocyte morphology, oxidative stress and histopathological lesions [3,4,5,6,7]. As well as in the evaluation methods and collected materials, there is much variation in the ages and inoculation routes, among which the shell membrane (SM), the allantoid fluid (AF), the chorioallantoic membrane (CAM), the yolk sac and the amnion are widely used [7,8,9,10,11].

Chicken embryo (CE) development is completed in just 21 days, being the fastest among the most used animal models and equal only to the mouse embryo [12]. Thus, on each day of incubation age, many physiological changes can significantly influence the studies and responses being evaluated, making the definition of the day of inoculation and collection crucial for realizing an excellent experimental study.

The standardization of parameters, such as the day of inoculation and collection, the inoculation route used, the biological materials collected, and the types of analyses performed are crucial for an experiment’s success. Studies that seek the standardization of such parameters are needed to better understand and identify the viability of CE and *in ovo* studies as models for preclinical tests. Thus, this study aims to verify if the results of viability, weight, pathological and histopathological changes, blood count, dosage of metabolites and/or enzymes and oxidative stress in CE of different ages are specific to the model, proposing and discussing the need for standardization for embryos, using as reference a virus that causes known changes in adult chickens and two drugs, DX which is a glucocorticoid widely used in animals and humans and can cause immunosuppressive effects and FG which is a stimulator of granulocyte precursors widely used in humans in treatment of cancer.

## 2. Materials and Methods

This research was performed in the following laboratories of the Uberlândia Federal University: Poultry Egg Incubation, Nanobiotechnology, Biochemistry, Molecular Biology, Animal Pathology, Veterinary Clinical, and Rodent Vivarium Network. The studies were divided into different phases. The project was evaluated and approved by the Ethics and Research with Animals Committee of the Universidade Federal de Uberlândia (certificate A011/20 and n° 008/21). All methods were performed in accordance with the CONCEA (Conselho Nacional de Controle de Experimentação Animal) guidelines and regulations, and the study is reported in accordance with ARRIVE guidelines.

### 2.1. Challenge of CE with Gammacoronavirus

The first part of the experiment was carried out in specific pathogen-free (SPF) eggs of *Gallus gallus* challenged with a *Gammacoronavirus*, the infectious bronchitis virus (IBV) that causes embryonic chicken lesions. The CE were incubated in an artificial incubator (Premium Ecológica^®^, Belo Horizonte, Brazil) at 37 °C, 58% humidity, with the eggs being turned at a two-hour interval. The eggs were weighed and inoculated at 10 embryonic incubation days (EID). This age was used as between 9–10 EID, lesions in CE can be distinguished [13]. A total of 4.35 log viral particles/CE/of IBV (strain Massachusetts, H52), diluted in sterile Phosphate-buffered saline (PBS), were inoculated in AF from 10 CE. In parallel, four embryos were inoculated as negative controls (NC) with sterile PBS (diluent of the virus). 

After 24 h of incubation, the CE were evaluated to remove the non-specific mortalities. At 17 EID, 0.5 mL of AF was collected. Blood was collected by sectioning the umbilical vessels and divided into two microtubes: one with EDTA K3 for blood cell analysis and the other with clot activator for serum analysis. In the AF and serum, the minerals, metabolites, and enzymes were quantified. After collecting the blood, the CE were immediately euthanized by decapitation, weighed and evaluated for macroscopic changes. The livers were collected, stored in formalin for histopathological tests and in liquid nitrogen for oxidative stress evaluation.

### 2.2. Inoculation with DX and FG

#### 2.2.1. Drug Dosages: Pilot Test

Prior to commencing the study, we carried out a pilot test to determine the dose of each drug. In this part of the experiment, we used a commercial line of CE, Hy-Line W36, as SPF birds only are mandatory for research with viruses. The eggs were incubated from zero EID in conditions identical to those mentioned in item 2.1. There is no standardized allometric extrapolation calculation for CE. Therefore, we usually test the adult chicken dose in 10 EID CE with an average weight of 20 g (CE and embryonic annexes). The intention was to determine the dose that would not kill the embryo. 

In the pilot test, we tested two routes: SM and CAM. We used an approximate dose for a hatched animal as the basis. The 2–4 mg/kg doses were used for DX [14]. According to the drug manufacturer, there is no indication of FG for birds, so we based it on human doses (5 mg/kg). Therefore, the following doses were tested: 4, 0.4 and 0.08 µg/CE of DX and 150, 15 and 1.5 µg/CE of FG, with three CE per dose via SM and CAM plus the NC for each route, totaling 42 embryos. After seven days, at 17 EID, the CE were evaluated for macroscopic changes. The dose chosen was based on the group with no expressive mortality or severe lesions in the embryos. For DX, the dose of 0.08 µg/CE did not cause gross lesions in any CE, but the 0.4 and 4 µg/CE resulted in green AF in CE inoculated via SM and CAM. Doses of 1.5, 15 and 150 µg/CE of FG did not alter the CE. Therefore, the used dose of DX was 0.08 µg/CE (~4 µg/kg) and FG was 150 µg/CE (~7.5 µg/kg). As the results for embryos inoculated via CAM and SM were identical, in the first moment, the SM route was used as it was the easiest, with a lower risk of death or lesions, and is an essential route for young or old embryos (which were used in this work).

#### 2.2.2. Drug Tests in CE

We started the experiment using CE at 12 EID (Hy-Line W36), as they are older CE and their organs and physiologies more mature than those of younger CE. The egg incubation conditions were identical to those mentioned in item 2.1. A total of 18 CE was used with six CE in the following groups inoculated via SM: (i) treated with 0.08 µg/CE of DX; (ii) treated with 150 µg/CE of FG; and (iii) NC, inoculated with water (diluent of drugs) only. The CE were treated at intervals of 24 h for three days. At 12 EID, the eggs were weighed. After seven days post-inoculation (pi), at 19 EID, we carried out blood and liver collection. We evaluated the weight and macroscopic lesions similar to those described in item 2.1, except for AF, as the amount of this fluid was very low at this age.

We also performed inoculations with FG (150 µg/CE) and DX (0.08 µg/CE) and a NC group at intervals of 24 h for three days, with six or eight eggs per group, at ages zero, three and seven EID via SM. After nine, 11 and 10 days of incubation, the CE inoculated at zero, three and seven EID, respectively, were euthanized by decapitation, weighed and evaluated regarding mortality and macroscopic lesions. We did not conduct blood, metabolites or microscopy analysis of the CE inoculated at zero, three and seven EID.

Another assay was performed in CE at 10 EID. In this part of the experiment, the route of inoculation for DX was SM. As there were no changes, as expected, in blood granulocyte counts in CE inoculated at 12 EID with FG, and there were no changes in young CE, the CAM route was used for FG in CE of 10 EID. As the FG pathway in humans is subcutaneous and the CAM pathway in CE has direct access to the vessels, we hypothesized that this could be an alternative to test this drug. Thus, it could be possible to assess whether the route could interfere with the results for FG. This part of the experiment was carried out twice, with a total of 18 CE each time, totaling 36 CE. The groups were divided as follows: (i) CE inoculated with 0.08 µg/CE of DX via SM; (ii) NC of DX–CE inoculated with water via SM; (iii) CE inoculated with 150 µg/CE via CAM; (iv) NC of FG–CE inoculated with water via CAM. The CE were treated at intervals of 24 h for 3 days. After seven days pi, at 17 EID, blood, AF, liver weight and macroscopic lesions were collected, similar to that described in item 2.1. The experiment was repeated at an interval of 15 days, identical to the previous experiment.

### 2.3. Weight of the CE and Annexes

During the experiments, the eggs were numbered and weighed on the first day of inoculation and their weights recorded. Then, the embryo and yolk were weighed immediately after collecting blood and AF on the day of collection. As the embryo weight is related to the initial egg weight, we performed an adjustment for an initial egg weight of 50 g, according to Ribeiro et al. (2020) [7].

### 2.4. Macroscopic Evaluations

We checked and counted the CE that died and identified the death date based on embryo development degree. For live animals, we observed their annexes for the presence of circulatory changes, malformation and colour changes. We also checked the embryos by external evaluation and evaluated the internal organs for circulatory changes, malformation and colour changes and compared the treated groups with the animals in their respective control group.

### 2.5. Blood Cell Count, Hematocrit, Hemoglobin and Erythrocyte Indices

To determine the hematocrit value (Ht), we filled capillary tubes up to 2/3 with a blood sample. We centrifuged them at 12,000× *g* for 5 min for later reading on a microhematocrit scale.

Hemoglobin (Hb) concentration was measured using the cyanmethemoglobin method based in Collier (1944) [15] with modifications using Drabkin’s solution and reading by spectrophotometry at an absorbance of 540 nm. The resulting colour is of an intensity proportional to the hemoglobin content in the blood. In parallel, we measured the hemoglobin by calculating 1/3 of the hematocrit value.

The total red blood cell (RBC) count was obtained using the Natt and Herrick (1952) [16] solution. Therefore, we performed a manual count in a Neubauer chamber for the hemacytometer method using blood dilution of 1:200. Counting was performed in the five diagonal squares of the central reticulum of the chamber. The result was multiplied by 10,000 to obtain the value of erythrocytes per microliter of blood [17]. 

The total white blood cell count and thrombocytes were also determined using the hemacytometer method by diluting the blood with the Natt and Herrick (1952) [16] solution. The leucocytes and thrombocytes were counted simultaneously in the four external reticula of the Neubauer chamber. The result was multiplied by 500 to obtain the value of leucocytes and thrombocytes per microliter of blood [17]. Two different people counted the cells to confirm the precision of the result. The characteristics of the cells (thrombocytes and eosinophils) were confirmed by cytochemistry.

We calculated the mean corpuscular volume (MCV), mean corpuscular hemoglobin (MCH) and mean corpuscular hemoglobin concentration (MCHC) using the formulas [18]:$$\mathrm{MCV}\;=\frac{\left(\mathrm{Hc}\;\times 10\right)\;}{\mathrm{RBC}}\;\mathrm{MCH}\;=\frac{\left(\mathrm{Hb}\;\times 10\right)}{\mathrm{RBC}}\;\mathrm{MCHC}\;=\frac{\left(\mathrm{Hb}\;\times 100\right)}{\mathrm{Hc}}$$

### 2.6. Characterisation of Blood Cells

The blood smear slides were prepared for staining with fast panoptic dye for differential leucocyte counts under an optical microscope (Olympus CX31, 100× immersion oil). Slides were also stained using the Periodic Acid Schiff (PAS) cytochemical technique for identifying thrombocytes and Sudan Black B (SBB) for identifying eosinophils [18]. Finally, the cells were counted twice at different moments under an optical microscope.

Knowing that serotonin binds to gaseous formaldehyde, we carried out the marking of thrombocytes. We based this on Swayne et al. (1986) [19] with some modifications. First, blood smear slides were placed in a box containing formalin and kept for 24 h at 50 °C. Next, we read the slides under a UV light microscope (EVOS FL Cell Imaging System, Life Technologies Corporation, Waltham, MA, USA). Thrombocyte diameter was measured using the ImageJ morphometry program. Slides from healthy and adult chickens (24 weeks of age) of the same line were prepared in parallel as a positive control.

### 2.7. Biochemical Analyses of the Serum and AF

As the collection of blood from the embryo is not simple and some researchers perform analysis of metabolites and enzymes in allantois, we performed the analysis in blood and allantois to determine if the results were similar. Biochemical analysis of serum and AF was performed in an automatic biochemical analyser (ChemWell^®^ 2910, Awareness Technology). The analytes aspartate aminotransferase (AST), alanine aminotransferase (ALT), gamma-glutamyl transferase (GGT), alkaline phosphatase (ALP), creatinine (Creat), uric acid (UA), calcium (Ca), phosphorus (P) and c-reactive protein (CRP) (Bioclin^®^, Minas Gerais, Brazil) were measured in both serum and AF. 

### 2.8. Oxidative Stress

Immediately after euthanasia by decapitation, the livers of the CE were removed and stored at −80 °C. The samples were homogenized with 10 mM sodium phosphate buffer (pH 7.4) and centrifuged at 800× *g*, at 4 °C for 10 min. The supernatant was used to quantify the biomarkers of oxidative stress.

#### 2.8.1. Reactive Oxygen Species (ROS)

The samples were incubated with dichloro-dihydro fluorescein diacetate (10 µM) and 5 mM Tris-HCl buffer (pH 7.4) for 3 min. Subsequently, the fluorescence was measured at 530 nm (excitation in 474 nm).

#### 2.8.2. Lipid Peroxidation

The liver homogenates were incubated with 0.67% thiobarbituric acid (TBA) and 10% trichloroacetic acid (TCA), for 120 min. Then, n-butanol was added to the samples to remove the organic phase and the fluorescence was measured at 553 nm, after excitation at 515 nm. Lipid peroxidation was determined using the malondialdehyde (MDA) analytical curve [20].

#### 2.8.3. Sulfhydryl Group

The sulfhydryl group was detected using ditionitrobenzoic acid (DTNB) diluted in 0.2 mM potassium phosphate buffer (pH 8.0). The liver homogenates were incubated for 30 min with 1 mM phosphate buffer (pH 7.4) and 10 mM DTNB solution. The presence of sulfhydryl groups was spectrophotometrically detected at 412 nm [21].

#### 2.8.4. Total Antioxidant Capacity

The liver homogenates were incubated with 300 mM sodium acetate buffer (pH 3.6), 10 mM 2,4,6-tri (2pyridyl)-striazine (TPTZ) and 20 mM ferric chloride at 37 °C for 6 min, at 593 nm. A trolox analytical curve determined the antioxidant capacity and sodium acetate buffer was used as a blank [22].

### 2.9. Histopathology

The fragments of liver fixed in 10% buffered formalin were processed for the preparation of histological slides stained with Hematoxylin and Eosin (HE) [23]. Histopathologic morphology was qualitatively assessed and classified for the presence of inflammation, degeneration, necrosis and circulatory change.

Intracellular fat deposits (lipidosis) were evaluated semiquantitatively and classified by assigning a score relative to the level of lipid deposits in the sample, according to the histological classification of Brunt et al. (1999) [24] modified by Angulo (2009) [25] with adaptations.

According to Fáncsi (1982) [26], the cytoplasm of hepatocytes from CE has drops of lipids, and the amount of lipid inclusions increases mainly around the twenty-seventh day. Thus, the presence of small vacuoles in the cytoplasm of hepatocytes was considered physiological in the assessment of lipidosis. The scores were evaluated according to the NC and divided into: 0 corresponding to normal, with discrete lipid deposits; 1 with slight lipid deposits; 2 with moderate lipid deposits; 3 with a marked level of lipid deposition.

In the analysis of the inflammatory process, the number of inflammatory foci and the number of inflammatory cells in the foci were microscopically evaluated in all the livers of the embryos according to the NC. The inflammatory process was classified as: 0 corresponding to normal, with no inflammatory process; 1—mild, when the number of inflammatory foci was less than three with mild inflammatory infiltrate; 2—moderate, when the number of inflammatory foci ranged from three to five with moderate inflammatory infiltrate; and 3—marked, when the number of inflammatory foci was greater than five with marked inflammatory infiltrate.

Regarding circulatory alterations, hemorrhage and congestion, they were evaluated semiquantitatively in relation to the NC as: 0 corresponding to normal, with no circulatory alterations; 1—light; 2—moderate; 3—accentuated.

### 2.10. Angiogenesis

Although no injury was observed in CE treated with FG at any dose or route and we noticed an increase in blood vessels in some eggs treated with FG. Therefore, we performed another experiment adding five and 25 times the dose of FG used in the previous investigation. The dose increase was to assess whether doses higher than those used in humans were capable of causing visible damage to embryos and test the hypothesis that FG induces angiogenesis. We tested 34 CE (Hy-Line W36) at three EID and divided them into the following groups: (i) treated with 0.08 µg/CE of DX; (ii) treated with 150 µg/CE of FG; (iii) treated with 750 µg/CE of FG; (iv) treated with 3.75 mg/CE of FG; (v) NC, inoculated with water (diluent of drugs) only, all by the SM route. The CE was treated at intervals of 24 h for days. At 12 EID, we evaluated the viability and macroscopic lesions. We evaluated the angiogenesis in CAM using ImageJ software with the Vessels Analysis [27] plugin.

### 2.11. Statistical Analysis

We assessed whether the data followed normality. For parametric data, we used Analysis of variance (ANOVA) followed by Tukey’s test and for non-parametric data, we used the Kruskal–Wallis test. For evaluation of two groups, we used the *t* test for parametric data and Kruskal–Wallis or Wilcoxon tests for non-parametric data. A Pearson correlation test was performed. We considered a 95% confidence interval using the program GraphPad Prism 9.2.

## 3. Results

### 3.1. The Virus Can Change Blood Cell Count, Calcium and Lipid Peroxidation in the Liver

Of the ten CE inoculated with *Gammacoronavirus*, three died, six had lesions and one survived without lesions. The injuries, as expected, were dwarf or curled embryos with a green liver in one CE, five tiny embryos (being 2 CE with a hemorrhagic or enlarged liver, and 1 CE with milky AF). The presence of the virus led to a change in the weight of live embryos, as expected. Embryos inoculated with the virus had an average weight of 13.41 g (±2.86), whereas the NC weighed 18.17 g (±1.62) with *p* = 0.0173. We could not weigh the yolk sac of embryos inoculated with the virus because they disintegrated during manipulation. We performed a blood cell count, quantification of metabolites, minerals and enzymes and the results are described in Table 1 and Table 2.

Figure 1 shows the levels of ROS, ferric reducing antioxidant power (FRAP), lipid peroxidation and sulfhydryl groups in the liver of CE infected with *Gammacoronavirus*. In comparison with the NC, the virus decreased the lipid peroxidation levels (*p* < 0.05) of the embryos. 

In the histopathological analysis of the liver, congestion and hemorrhage were observed in CE infected with the virus. There was also a moderate inflammatory process predominantly composed of heterophils (Table 3 and Figure 2).

### 3.2. The Age of Inoculation and Type of Drug Result in Different Effects on Injury, Mortality or Embryo Viability

The CE were inoculated with the same dose of drugs at 0, 3, 7, 10 and 12 EID for three days. After inoculation, the CE were evaluated daily by light candling, and the dead CE necropsied and removed. Between 7–12 days of inoculation, the eggs were opened and necropsied (Table 4).

Of the dead embryos inoculated at zero EID with DX, two were hemorrhagic and died between three to four EID; two died at nine EID and had malformed heads, and two survived but had malformed heads. The only dead CE treated with FG died at nine EID without a specific injury. At three EID, the CE inoculated with DX had the CAM not wholly formed around the CE, and the yolk sac leaked very quickly. Dead CE inoculated at seven EID with DX presented a smaller size, gelatinous albumen, irregular warping and a green liver, and died between 14–15 EID. Injured live CE showed decreased size and curling, plus petechiae on top of the head. The injured CE in NC and FG inoculated at 10 EID showed a delicate increase in the liver. The injured CE inoculated at 10 EID treated with DX had an enlarged liver which was dark or light green. The lesions observed in live injured CE inoculated at 12 EID with DX were white material over the heart (with UA aspect) and beige material within the allantois. This material was different in appearance and colour from the UA commonly found in the allantois in normal embryos.

### 3.3. The Weight of CE Can Be a Drug Analysis Tool Depending on Age

As shown above, most CE inoculated with DX at zero EID did not survive; thus, the weights of the embryos from the control group and the FG-treated group were evaluated, and there was no difference in weights. CE inoculated with DX at three EID showed decreased weight compared with NC and FG groups. The same occurred in CE inoculated at seven EID. The yolk of embryos inoculated with DX presented a lower weight in relation to the NC and FG groups inoculated at three and seven EID. In CE inoculated at 10 EID and 12 EID, a lower weight of embryos treated with DX compared with embryos from NC and FG groups was observed, but this difference was not observed in the yolk weights of these animals (Figure 3).

### 3.4. The Blood Cell Count Can Change According to the Age of Inoculation

The blood samples collected from CE treated with FG did not show a statistical difference compared with the NC group for inoculations made at 10 EID and 12 EID, but the same did not occur for CE treated with DX. Compared with the NC group, the samples from embryos inoculated with DX at 10 EID had a smaller number of erythrocytes, lower values of hematocrit and hemoglobin, and a smaller number of thrombocytes (Table 5). No statistically relevant differences were observed in the amounts of total leucocytes. In CE inoculated at 12 EID, embryos treated with DX showed a lower number of erythrocytes. The number of thrombocytes was more significant than the NC group, contrary to the CE treated at 10 EID. In the group inoculated at 12 EID, there was a statistical difference for the amounts of total leucocytes, which was higher for the group treated with DX, heterophiles which was higher, and lymphocytes which were lower compared with the NC group (Table 6).

### 3.5. Not All Metabolites, Enzymes and/or Minerals Change Even in Liver-Damaged Embryos

Among the serum and AF samples analyzed only AST from serum samples of CE treated with DX at 10 EID showed a statistical difference compared with the NC group (Table 7, Table 8 and Table 9).

### 3.6. The Amount of Metabolites, Enzymes and Minerals in Serum and Allantois Are Not Always Identical

The UA, Creat, ALP, GGT, ALT, AST, Ca and P of serum and AF samples from embryos treated with FG and DX at 10 EID were evaluated. Only AST and calcium in CE treated with FG and Crea, ALP and P in CE treated with DX showed similar values for the two analyzed samples (Table 7 and Table 8). Although the values of the metabolites and enzymes were not equivalent in all analyses, the results were identical since there were no statistical differences except for AST, which increased in the group treated with DX at 10 EID in serum but not in the AF.

### 3.7. The Age of Inoculation Is an Essential Factor for Changing Oxidative Stress Parameters

Figure 4 shows the levels of sulfhydryl groups, FRAP, lipid peroxidation and ROS in the liver of CE inoculated with FG and DX at 10 and 12 EID. In comparison to the NC, FG increased the levels of sulfhydryl groups (*p* < 0.05) of CE treated at 10 EID. At the same EID, embryos treated with DX had increased levels of ROS production. In CE treated at 12 EID with FG, there was no difference between the NC group, but the CE treated with DX showed a decrease in the sulfhydryl group and FRAP levels and an increase in the levels of lipid peroxidation and ROS production.

### 3.8. There Is a Correlation between the Results for Haemoglobin Using a Drabkin Solution or Performing the Calculation of One Third of the Haematocrit

There was a correlation between the assessment of hemoglobin by calculating one third of the hematocrit and the method using the Drabkin solution. The r value in the group of CE inoculated at 12 EID was 0.61, representing a moderate correlation. In the groups treated at 10 EID with FG, DX and virus, the r value was 0.86, 0.85 and 0.86, respectively, showing a strong correlation.

### 3.9. Characterisation of Granulocytes and Thrombocytes by Cytochemistry

Thrombocytes were labelled by the formaldehyde method (Figure 5) and the analysis showed that the thrombocytes’ diameters in CE were lower than in adult animals. Moreover, the FG and DX increased the diameter of the thrombocytes.

Blood smear slides were stained by PAS cytochemical methods for identification of thrombocytes by staining of cytoplasmic glycogen granules and SBB for identification of eosinophils by staining of cytoplasmic granules. These two methods were used to differentiate lymphocytes that were negative for PAS and heterophils that were negative for SBB (Figure 6).

### 3.10. Unidentified Granulocytes Were Found in Several Groups

Despite the various methods of analysis used to identify the cells, some cells with a rounded nucleus and basophilic round cytoplasmic granules were found on some slides in groups NC, FG, DX, and in the group infected with the virus (Figure 7). These cells were not positive by cytochemistry used in our work.

### 3.11. Liver Histopathological Analysis

The degenerative change observed in the CE livers was lipidosis. The hepatocytes were vacuolated, ranging from few and small light vacuoles to large light vacuoles, shifting the nucleus to the periphery, with the hepatocyte resembling an adipocyte. As in virus-infected embryos, the inflammatory process had a periportal distribution and was predominantly composed of heterophils (Figure 8).

The animals inoculated with FG at 10 EID showed an inflammatory process with infiltration of heterophils. Embryos inoculated with DX at 10 EID showed a mild inflammatory process and circulatory alteration and animals inoculated at 12 EID showed a mild inflammatory process and degeneration (lipidosis) (Table 10).

### 3.12. FG in High Doses Does Not Cause Injury to Angiogenesis in CE

We tested the hypothesis that FG could cause angiogenesis in CE using several doses. Unlike that believed, FG did not induce angiogenesis (measured by vascular density and vessel length density), just as DX did not change the number of vessels (Appendix A). As already reported in the previous experiment, DX led to incomplete CAM formation. The high dose of FG also did not cause visible changes in the embryo.

## 4. Discussion

CE provide an ideal model for investigating the development of pathogens such as viruses and testing drug effects, evaluating both toxicity and pharmacokinetics. Thus, CE as an animal model can be an important ally for carrying out experiments that require quick and less complex execution and with limited resources.

The CE and its annexes constitute a favorable environment for replicating several viruses, being widely used for the isolation or production of vaccines [9]. IBV causes dwarfism, hemorrhage and death when inoculated into embryonated eggs. Some strains of the *Gammacoronavirus* can cause nephropathies and others hemagglutination [28]. However, IBV has a minimal agglutination capacity for erythrocytes in chickens [29]. Our study showed a decrease in hemoglobin concentration, hematocrit value and the number of erythrocytes in infected embryos compared with the NC group, indicating anemia. Hematimetric indices did not differ in relation to the NC group. Thus, we can classify anemia as normocytic and normochromic. This means that the cells were of standard size and hemoglobin concentration, thus there was a decrease in RBC, but there was no response from the bone marrow to release cells into the bloodstream. Normocytic and normochromic anemia can indicate decreased erythrocyte production, which can develop rapidly in birds with diseases involving infectious agents [18].

Although there was no statistical difference in the number of leucocytes in the infected animals compared with embryos in the NC group, some animals had a high leucocyte count (>30,000/mm^3^). In contrast, others had a low count (<4000/mm^3^) (Appendix A). These results contributed to the increase in the standard deviation in this analysis. During an infection the response of the white blood cell count can vary. In the initial of disease, leucocytosis may occur as a response by the body. However, as the leucocytes are consumed, leucopenia may occur when the demand for leucocytes is above the production capacity of these cells by the bone marrow [18]. From 12 EID, the hematopoiesis in CE is more active and occurs mainly in bone marrow [30]. Therefore, in this work, hematopoiesis in the CE was intense since we used animals between 10–17 EID. However, although the CE used in this work were from the same flock and the mother had high consanguinity, there may be an inherent variation in the model itself.

There was no change in the Creat and UA levels in serum and allantois samples from infected CE compared with the NC group, and macroscopic kidney lesions were not observed either. On the other hand, the liver of infected animals was macroscopically altered, with an increase in volume and a greenish colour. In addition, we observed a moderate inflammatory reaction by histopathologic analyses (Figure 2), although liver enzymes were not increased in the serum or allantois compared with the control group [31]. There was also a decrease in lipid peroxidation rather than an increase. The absence of a rise in GGT, ALT, AST and the reduction in lipid peroxidation indicate that the stress was very intense, reducing liver responsiveness.

Serum calcium levels from virus-infected CE decreased compared with the NC group. Possibly the viruses can hijack the host cell’s machinery and utilize the host cell’s calcium to create an environment adapted to meet their own demands for replication [32]. This hypothesis is supported in the work of Cao et al. (2011) [33]. These authors observed that the expression levels of some calcium-binding proteins were increased after *in ovo* IBV infection. These proteins facilitate the transcellular transport of calcium, suggesting that IBV may disrupt cellular Ca^2+^ homeostasis for its own benefit.

In addition to virus experiments, drug studies using CE as a model provide a technically simple way to study complex biological systems for in vivo drug toxicity and pharmacokinetic assessment. For example, DX is a widely used glucocorticoid. Its administration in CE can cause immunosuppressive effects increasing embryonic catecholamines that alter development and cause death [34].

There was high macroscopic lesion and mortality rate after the DX inoculation, similar to other studies [34,35,36]. As expected, the mortality was age-dependent. In addition, in our study, CE treated with DX reduced the weight in all CE that lived. The reduced weight in CE treated with DX can be related to muscle and bone development inhibition [35,36]. Furthermore, high doses of glucocorticoids can generate suppression of growth hormone activity in the pituitary gland which is fully established at the beginning of the last week of the embryonic development of CE [36]. Thus, we can suggest that DX promotes a delay in embryonic development, negatively influencing embryo weight gain, which may explain the delay in the complete formation of CAM found by us.

Unlike DX, FG did not cause significant changes and deaths. The only death observed in an embryo inoculated at zero EID was not accompanied by changes and occurred in an embryo from the NC group, probably caused by a natural process. Similarly, the injury in the CE inoculated via CAM possibly resulted from the inoculation process since hemorrhage may occur using this route.

CE treated with DX at 10 and 12 EID were anemic. The CE treated with DX showed a decreased erythrocyte count and decreased hematocrit and at 10 EID, hemoglobin concentration compared with CE in the NC group. There were also decreases in hematocrit, hemoglobin and erythrocyte values in animals treated with DX at 12 EID, but only erythrocytes showed a statistical difference for the NC group. Despite that, there was a high correlation between hemoglobin and hematocrit (r value = 0.78), hematocrit and erythrocyte (r value = 0.77) and hemoglobin and erythrocytes (r value = 0.81). Therefore, the erythrocyte reduction accompanied the reduction in hemoglobin and hematocrit. Although the CE treated with DX had anemia, the hematimetric indices were not statistically different between the groups treated and the NC, defining anemia as normocytic and normochromic. Chickens treated with corticosteroids show increased energy expenditure [37]. Embryos treated with DX at 10 EID had higher yolk consumption, indicated by the lower yolk weight of these embryos compared with animals in the NC group. The nutritional deficit associated with liver damage caused by the drug may have impaired the production of erythrocytes, causing normocytic normochromic anemia.

In CE inoculated with DX at 10 EID, there was a decrease in the number of thrombocytes compared with embryos from the NC group. Perhaps the production of thrombocytes in CE treated with DX at 10 EID was impaired by the damage caused in the liver. Thrombopoietin (TPO) is the regulator of megakaryocyte development and thrombocyte production and its expression in chickens occurs mainly in the liver [38]. Hemopoietic activity in the liver starts at seven EID with a peak at 14 EID [39], but is more active from 12 EID [30]. In animals treated with DX at 12 EID, the opposite was observed. There was an increase in the number of thrombocytes compared with CE from the NC group. Thrombocytosis may reflect a rebound response after recovery from other conditions associated with excessive use of thrombocytes. Considering that CE from 12 EID had a more mature liver, active bone marrow and hemopoietic activity [30], thrombocytosis can be explained in CE treated at 12 EID. Additionally, it should also be considered that in birds, thrombocytes have a phagocytic function and the influence of glucocorticoids on these cells is unknown [40].

In embryos treated with DX at 12 EID, there was an increase in the number of leucocytes, different to that observed in CE inoculated at 10 EID. This can be explained by the onset of lymphoid activity in the Bursa of Fabricius at 12 EID [41], with a greater capacity to respond to the stimulus caused by the drug. In mammals treated with DX, an initial leucocytosis may occur, mainly due to neutrophilia [29]. In fact, CE inoculated at 12 EID showed an increase in H/L ratio due to the increase in heterophils and a decrease in lymphocytes. This corroborates several studies in born animals [42,43,44]. During CE development, granulopoiesis is more predominant; however, at hatching, the granulocytes begin to be replaced by lymphocytes within first three days [18].

Taken together, the results show that embryos with a small difference in embryo development stage can completely alter the cell count response. We do not know if this occurs with other drugs. However, knowing that the embryo is an ascending animal model, further work must be carried out to consider the best age for using the model depending on the expected objective.

As a granulocyte colony-stimulating factor (G-CSF), FG is used in human medicine to increase levels of neutrophils in the bloodstream. Therefore, we expected that the same effect would be observed in the CE in this experiment, increasing heterophils that have characteristics and performance corresponding to human neutrophils. However, no increase in granulocytes were observed in animals treated with FG at 10 EID or 12 EID either by inoculation in CAM or SM. Perhaps this did not occur because FG is a synthetic compound for human use and may not have the same results in other species.

The analysis of biochemical parameters provides essential data for the assessment of the clinical status of the animal. However, blood samples collected in research with CE do not always allow this analysis because it is not easy and takes time. Therefore, alternatives samples have been used to assess these parameters, such as amniotic fluid and AF [5,7,45]. In this work, we compared the biochemical analyses of serum and AF and observed that the values found for the two samples were not always similar. However, the differences observed between groups in serum samples were also observed in AF samples, except for AST, which may indicate that this enzyme does not have a good analysis from allantois.

It is known that proteins found in serum have different physical and biochemical properties and change in various physiological and pathological conditions. One of the problems with enzymatic analysis methods is that the reagents were designed to provide the substrate and its optimal concentrations for human plasma, but these variables can change depending on the species. For example, birds have deficient levels of activity of the enzyme ALT in the liver tissue, so in cases of severe liver damage, this enzyme may present normal values. AST activity occurs in multiple tissues, but the main ones are liver and muscle, being considered sensitive but not very specific in cases of liver problems. GGT activity is increased in all conditions in which hepatocellular damage is present. ALP is associated with the regulation of bird growth, participating in chondrogenic and osteoblastic activities. Thus, physiological variations can be observed, with higher activity levels resulting from bone growth in young birds. Elevations in ALP may be associated with liver disease even if its activity in this organ is minimal [26].

In our study, only the AST enzyme of CE treated with DX at 10 EID showed a statistical difference compared with the CN group. However, histopathological findings and results of oxidative stress biomarkers indicate that there was liver damage. At 12 EID, there was no increase in AST in CE treated with DX. At this age, just 50% of the CE had a macroscopic injury (Table 3) and maybe because of this, the AST level did not increase. However, the AST maximum value of the group treated of DX at 19 EID was greater than the NC (Appendix A). Thus, AST may be the best parameter for analysis of liver function in embryo serum.

High concentrations (up to five times) of UA in plasma can lead to precipitation of this acid in the form of crystals, which accumulate in tissues. Situations of hypouricemia are rare and may be related to severe liver damage with a consequent decrease in UA production [46]. The excretion of Crea occurs via the kidneys, but in birds, most creatine is excreted before being converted to creatinine [18]. Thus, increased Crea concentrations are rare and may occur in severe renal impairment, significantly if filtration is affected [46]. In our study, no changes in urea or Creat were observed in any of the treatments. Macroscopic lesions were also not observed. This can mean that there was no damage to the CE kidneys. However, as we did not perform histopathological analysis of the kidneys, we cannot rule out the possibility that the high damage caused by the treatment does not lead to changes in biochemical parameters.

Drugs can cause increased production of oxidants and the formation of free radicals, which, by exceeding the body’s ability to neutralise and scavenge these radicals, can cause organ damage [47]. Our study showed an increase in oxidative stress biomarkers ROS in the livers of embryos treated with DX at 10 EID and 12 EID and an increase in lipid peroxidation in embryos treated at 12 EID. In adult animals, corticosteroids can increase oxidative stress [48], but a similar approach was never studied in CE. The increased energy expenditure triggered by high circulating corticosteroid levels may be responsible for the increased of ROS as reflected by increased lipid peroxidation [48]. The body uses enzymatic and non-enzymatic antioxidants to neutralize damage caused by free radicals and minimize excessive oxidative stress. Sulfhydryl groups are an example of essential antioxidants status, suggesting how this process is controlled and protected from oxidative damage. Embryos that were treated at 12 EID showed decreased sulfhydryl groups and FRAP values, indicating that these embryos had difficulty matching the damage caused by free radicals in the induction of oxidative stress by DX.

Embryos treated with FG at 12 EID did not show changes in oxidative stress biomarkers compared with CE from the NC group. However, in CE inoculated at 10 EID via CAM, FG increase the sulfhydryl group content. Filgrastim is a granulocytic colony-stimulating factor (G-CSF). It has biological activity identical to that of endogenous human G-CSF with a free cysteine at position 17 with an ionized sulfhydryl group that is very reactive to free radical oxidation [49]. From this, we can conclude that FG may have shown an antioxidant effect, with possible protection of the embryo against ROS.

To measure hemoglobin in this experiment, we used Drabkin’s solution and compared it with the values obtained using the 3Hb/Hc ratio. Our study showed a moderate correlation in embryos treated at 12 EID and a strong correlation in embryos treated at 10 EID [50]. Thus, we can conclude that the calculation based on hematocrit can be used to approximate the hemoglobin value in situations where measurement by spectrophotometry is not possible. However, this replacement is only possible if there is no suspicion of hemolysis, since hemolysis due to collection or pathological problems, promotes a decrease in hematocrit without a proportional reduction in hemoglobin.

To perform the differential count of leucocytes in birds, there is a great difficulty is differentiating between thrombocytes and lymphocytes. Though not identical, the nuclei of thrombocytes and small lymphocytes are too similar to serve as a basis for distinguishing between these two cell types. In the present study, cells in fast panoptic stained CE blood smears with small, round or oval nuclei with dense chromatin clumps were categorized as thrombocytes if they had cytoplasmic vacuoles and colourless cytoplasm. Cells classified as small lymphocytes had similar nuclei but scant amounts of blue or dark blue cytoplasm without vacuoles. To establish a basis for the categorization of these cells, some cytochemical properties of these cells were compared. The typical thrombocytes were visualized by UV after being exposed to gaseous formaldehyde. Swayne et al. (1986) [19] observed that most of the thrombocytes (99%) had been fluorescent after gaseous formaldehyde treatment and all of the small lymphocytes had been non-fluorescent. This fluorescence resulted from serotonin condensation products. We measured the thrombocytes found in the slides of animals inoculated in this study and compared them with cells found in the blood smear slides of adult chickens. Thrombocytes from adult chickens had a larger size compared with the NC group, which averaged 8.5 µm. We could not determine why FG increased the size of thrombocytes.

Using the PAS, we characterized the thrombocytes as PAS positive, whereas lymphocytes and erythroblasts were PAS negative (Figure 6). The eosinophils were SBB positive, whereas the heterophils were SBB negative (Figure 6). The use of PAS and SBB is important for better cell classification.

Despite the different forms of differentiation used to identify cells in the blood smears of embryos in this experiment, some cells remained without conclusive identification. The authors identified these cells as being a type of granulocyte, but they could be confused with eosinophils, basophils or granulocyte precursors [17,18,51]. These cells were found in samples from embryos treated with FG (20%) and DX (25%) inoculated at 10 EID, animals infected with virus (100%), and in animals from the NC group (23.07%) inoculated at 10 EID.

Another response observed in the use of FG in humans is the stimulation of endothelial cells with consequent angiogenesis [52,53,54]. In this study, we analyzed the blood vessels of the CAM of animals inoculated with FG to identify a possible increase in vessel density characterizing the occurrence of angiogenesis. However, there was no statistical difference between CE inoculated with FG and the NC group.

## 5. Conclusions

Overall, the results of this work provide vital information on the use of CE as an experimental model. The response of CE to challenges from viruses and drugs does not always go as expected. Although macro and microscopic damage were visible in viruses, white blood cell counts and inflammation biomarkers such as CRP did not change. It is important to mention that some drugs can be innocuous and not result in expected effects on CE, which was the case with FG. In the case of DX, changes in blood parameters and biomarkers seem to be inherent to the model and are highly dependent on the developmental stage of the CE.

This article reiterates the wonderful value of the CE as an animal model. However, our work sheds light on the importance of standardization and the correct use of the model (considering the laboratory analysis, drug, age, route) so that the infection, toxicity and pharmacokinetic results are reliable.

## Figures and Tables

**Figure 1 animals-12-01156-f001:**
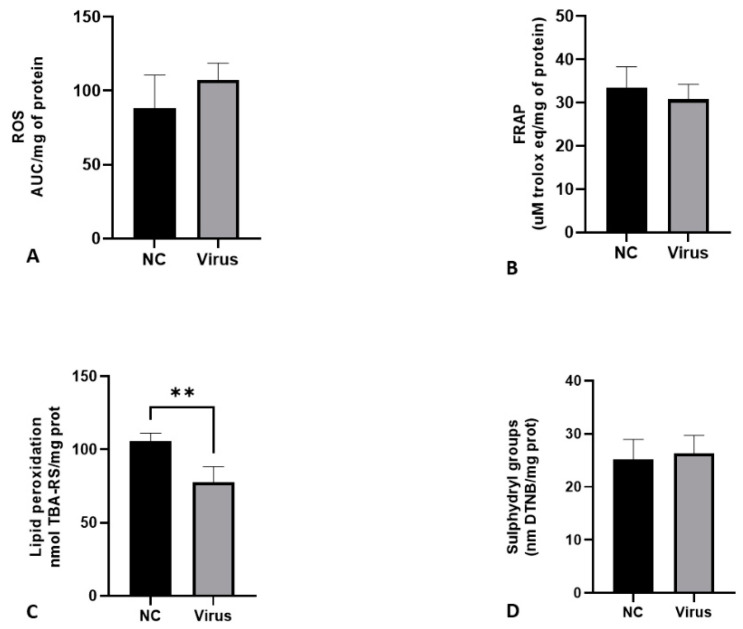
Levels of oxidative stress biomarkers in terms of ROS production (**A**), FRAP (**B**), lipid peroxidation (**C**) and sulfhydryl groups (**D**) in liver of CE after infection with *Gammacoronavirus*. The asterisk symbol indicates a statistical difference between the groups (** = *p* ≤ 0.01). FRAP: ferric reducing antioxidant power. ROS: reactive oxygen species. NC: Negative control.

**Figure 2 animals-12-01156-f002:**
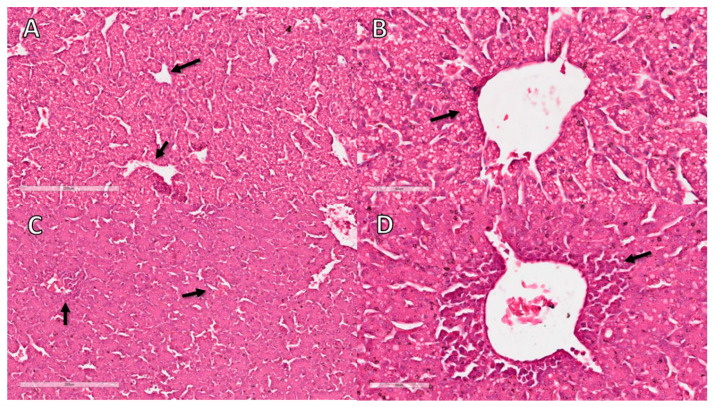
Histopathological analysis of CE liver infected with *Gammacoronavirus* at 10 EID. (**A**) NC sample showing sinusoids with a minimal number of erythrocytes (arrow). (**B**) NC sample without perivascular inflammatory infiltrates (arrow). (**C**) Liver of virus-infected CE with congestion characterised by erythrocyte-filled sinusoids (arrows). (**D**) Liver of virus-infected CE with inflammatory infiltrate (arrow). The bars in images (**A**,**C**) represent 200 µm and the bars in images (**B**,**D**) represent 60 µm.

**Figure 3 animals-12-01156-f003:**
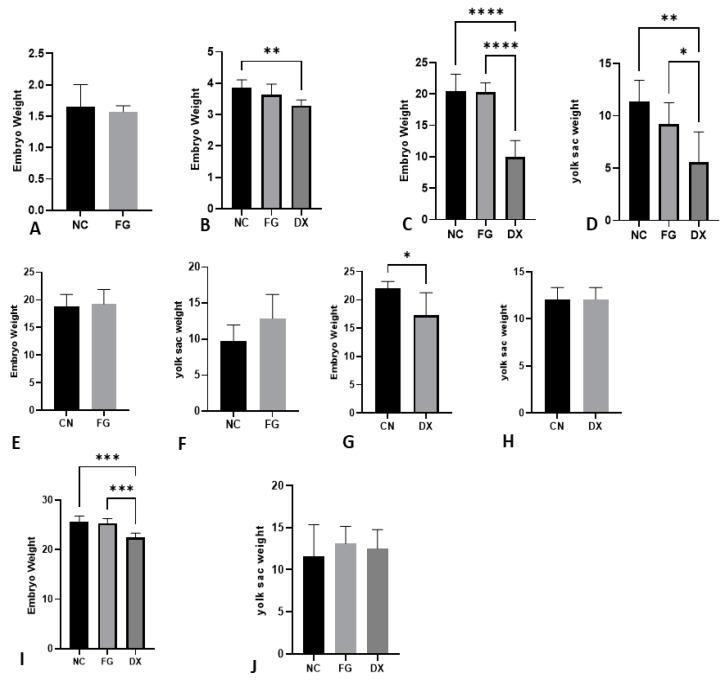
Adjusted embryo and yolk weight in grams at different ages and treatments. Embryo weight inoculated at 0 EID (**A**), 3 EID (**B**), 7 EID (**C**) inoculated via SM, 10 EID inoculated with FG via CAM (**E**), 10 EID inoculated with DX via SM (**G**), 12 EID inoculated at SH (**I**). Yolk sac weight at 7 EID (**D**), 10 EID inoculated with FG via CAM (**F**), 10 EID inoculated with DX (**H**) via SM, 12 EID (**J**) inoculated at SH. In CE inoculated at 0 EID, the CE of the DX group died. We could not weigh the yolk sac of embryos inoculated with the DX at 0 and 3 EID because they disintegrated during manipulation. The asterisks symbols indicate a statistical difference between the groups (* = *p* ≤ 0.05; ** = *p* ≤ 0.01; *** = *p* ≤ 0.001; **** = *p* ≤ 0.0001). NC: Negative control. FG: Filgrastim. DX: Dexamethasone.

**Figure 4 animals-12-01156-f004:**
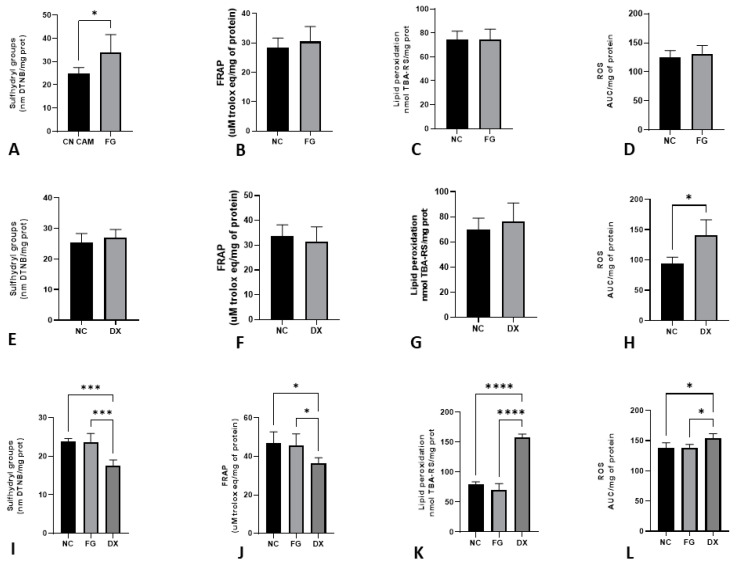
Levels of oxidative stress biomarkers in terms of sulfhydryl groups (**A**), FRAP (**B**), lipid peroxidation (**C**) and ROS production (**D**) in chicken embryos’ livers after inoculation with FG via CAM at 10 EID; sulfhydryl groups (**E**), FRAP (**F**), lipid peroxidation (**G**) and ROS production (**H**) in chicken embryos’ livers after inoculation with DX via SM at 10 EID; sulfhydryl groups (**I**), FRAP (**J**), lipid peroxidation (**K**) and ROS production (**L**) in chicken embryos’ livers after inoculation with FG and DX via SM in 12 EID. The asterisks symbols indicate a statistical difference between the groups (* = *p* ≤ 0.05; *** = *p* ≤ 0.001; **** = *p* ≤ 0.0001). FRAP: ferric reducing antioxidant power. ROS: Reactive oxygen species. NC: Negative control. FG: Filgrastim. DX: Dexamethasone.

**Figure 5 animals-12-01156-f005:**
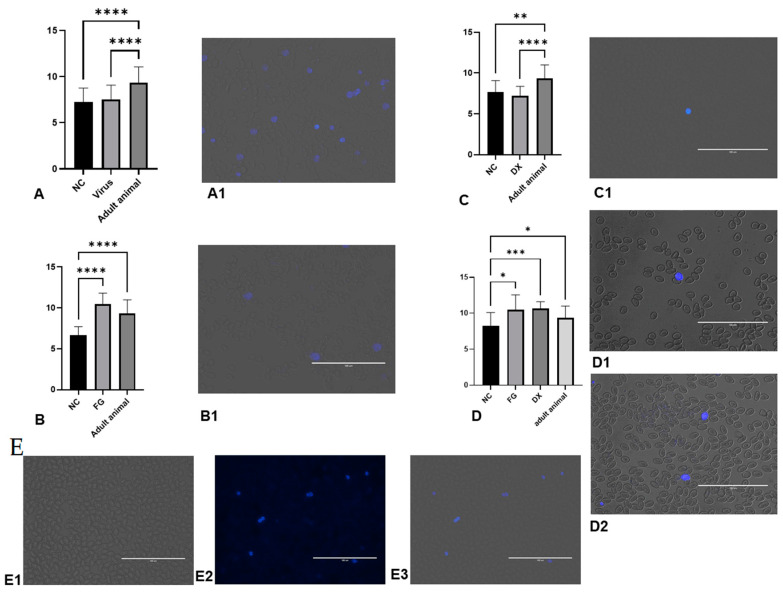
Diameter of formalin-labelled thrombocytes evaluated. (**A**) diameter of thrombocytes in SPF CE inoculated with *Gammacoronavirus*. (**A1**) thrombocytes stained in group treated with virus. (**B**) Diameter of thrombocytes in CE treated with filgrastim at 10 EID. (**B1**) thrombocytes stained in group treated with FG inoculated at 10 EID via CAM. (**C**) Diameter of thrombocytes in CE treated with DX inoculated at 10 EID via SM. (**C1**) thrombocytes stained in the group treated with DX. (**D**) Diameter of thrombocytes in CE treated with FG and DX inoculated at 12 EID via SM. (**D1**) thrombocytes stained in the group treated with FG. (**D2**) thrombocytes stained in the group treated with DX. (**E**) Thrombocytes stained in the NC group in CE inoculated at 10 EID. (**E1**) trans, (**E2**) UV, (**E3**) over. The asterisks symbols indicate a statistical difference between the groups (* = *p* ≤ 0.05; ** = *p* ≤ 0.01; *** = *p* ≤ 0.001; **** = *p* ≤ 0.0001). NC: Negative control. Scale bar = 100 µm.

**Figure 6 animals-12-01156-f006:**
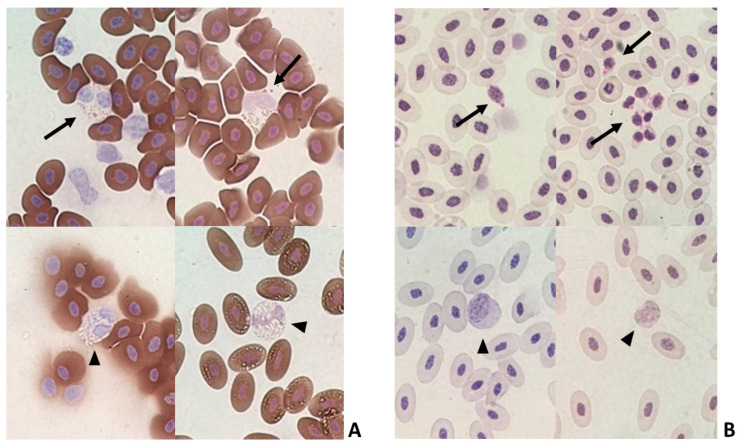
Blood smear slides stained by cytochemical methods. (**A**) Sudan Black B (SBB) positive eosinophils (arrow) and SBB negative heterophils (arrowhead). (**B**) Periodic Acid Schiff (PAS) positive thrombocytes (arrow) and PAS negative lymphocytes (arrowhead).

**Figure 7 animals-12-01156-f007:**
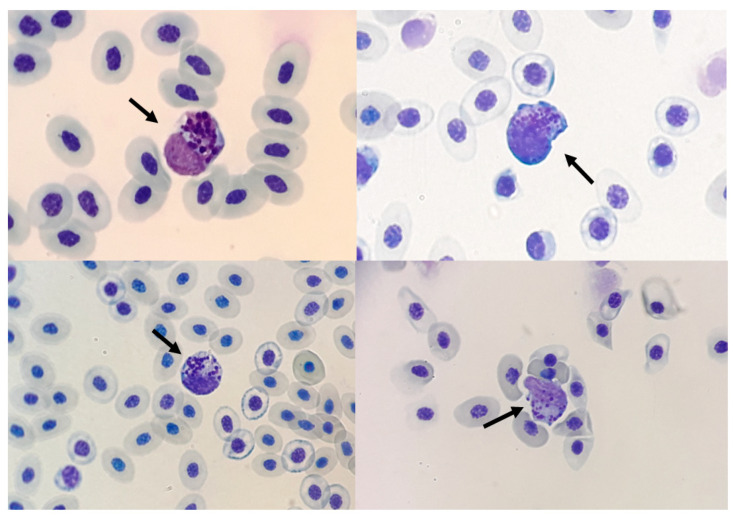
Unidentified granulocytes in blood smear stained with fast Panotico^®^ (arrow 100×).

**Figure 8 animals-12-01156-f008:**
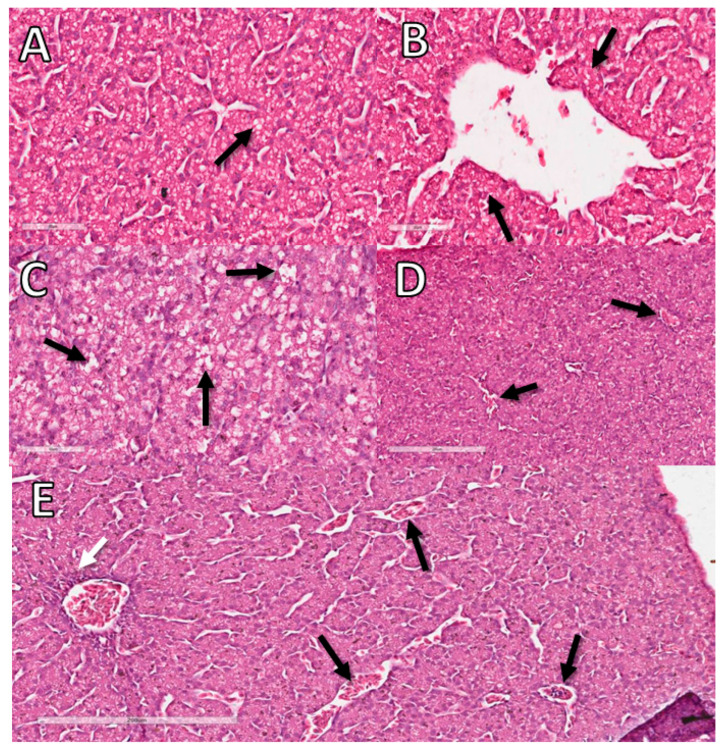
Histopathological analysis of the livers of CE treated with DX and FG at 10 EID. (**A**) CE liver sample from NC with a small amount of lipids in vacuoles in hepatocytes considered normal for comparison with the other groups (arrow). (**B**) Sample from the NC without inflammatory infiltrate (arrow). (**C**) DX-treated CE sample showing large cytoplasmic vacuoles in hepatocytes containing lipid characterizing lipid degeneration (arrow). (**D**) DX-treated CE sample with sinusoids filled with erythrocytes characterizing congestion (arrows). (**E**) FG-treated CE sample with perivascular inflammatory infiltrate (white arrow) and congestion (black arrows). The bars in images (**A**–**C**) represent 60 µm and the bars in images (**D**,**E**) represent 200 µm.

**Table 1 animals-12-01156-t001:** Blood cell counts in CE treated with *Gammacoronavirus* inoculated at 10 EID.

	NC	Virus
Ht (%)	21.00 (±1.63) ^a^	15.60 (±2.30) ^b^
Hg (g/dL) (1/3 Ht)	7.00 (±0.55) ^a^	5.20 (±0.77) ^b^
Hg (g/dL) (Cyanmethemoglobin)	8.05 (±0.16) ^a^	6.23 (±0.99) ^b^
Erythrocytes × 10^6^/mm^3^	2.05 (±0.15) ^a^	1.59 (±0.23) ^b^
MCH (pg)	35.15 (±3.47) ^a^	40.08 (±9.95) ^a^
MCV (fL)	38.15 (±3.47) ^a^	40.08 (±9.95) ^a^
MCHC (g/dL)	96.96 (±17.07) ^a^	100.6 (±27.12) ^a^
Thrombocytes × 10^4^/mm^3^	2.21 (±0.51) ^a^	2.75 (±2.05) ^a^
Leucocytes × 10^3^/mm^3^	8 (±2.67) ^a^	20.4(±22.5) ^a^
Monocytes/mm^3^	28 (±55) ^a^	254 (±465.5) ^a^
Lymphocytes/mm^3^	650 (±730) ^a^	2170 (±2943) ^a^
Heterophiles × 10^3^/mm^3^	7.29 (±2.27) ^a^	17.6 (±18.8) ^a^
Heterophile/lymphocyte	24.09 (±18.53) ^a^	28.32 (±20.51) ^a^
Eosinophils/mm^3^	0.00 (±0.00) ^a^	0.00 (±0.00) ^a^
Unidentified granulocytes/mm^3^	23.75(±47.5) ^a^	259 (±459.1) ^a^
Basophils/mm^3^	0.00 (±0.00) ^a^	0.00 (±0.00) ^a^

The value in parentheses is the standard deviation. Different letters on the same line show statistical difference (*p* < 0.05). Monocytes, MCV: non-parametric test. Unidentified granulocytes: cells not tagged in Sudan Black B or PAS but not having a standard format. NC: Negative control. Ht: Hematocrit. Hg: Hemoglobin. MCH: corpuscular volume. MCH: mean corpuscular hemoglobin. MCHC: mean corpuscular hemoglobin concentration.

**Table 2 animals-12-01156-t002:** Quantification of metabolites, minerals and enzymes in CE treated with *Gammacoronavirus* inoculated at 10 EID.

	NC (Serum)	Virus (Serum)	NC (AF)	Virus (AF)
UA (mg/dL)	21.76 (±23.13) ^a^	22.63 (±15.17) ^a^	18.53 (±21.53) ^a^	54.30 (±29.89) ^a^
Creat (mg/dL)	1.42 (±1.21) ^ab^	0.50 (±0.35) ^a^	3.21 (±1.41) ^b^	1.55 (±1.11) ^ab^
ALP (U/L)	1857 (±1108) ^a^	2258 (±1540) ^a^	80.40 (±26.37) ^b^	55.74 (±49.27) ^b^
GGT (U/L)	81.53 (±55.34) ^a^	252.4 (±121.5) ^a^	210.00 (±172.2) ^a^	184.00 (±119.40) ^a^
AST (U/L)	264.00 (±40.84) ^a^	451.00 (±281.1) ^a^	200.00 (±123.3) ^a^	592.00 (±283.10) ^a^
ALT (U/L)	57.50 (±233.6) ^a^	92.00 (±85.73) ^a^	63.00 (±38.1) ^a^	48.00 (±22.80) ^a^
CRP (mg/L)	24.55 (±10.31) ^a^	24.75 (±9.91) ^a^	59.50 (±32.55) ^a^	84.00 (±45.72) ^a^
Ca (mg/dL)	95.23 (±66.18) ^a^	10.23 (±8.08) ^b^	16.05 (±10.48) ^b^	16.60 (±13.77) ^b^
P (mg/dL)	6.82 (±3.70) ^a^	5.50 (±2.96) ^a^	19.19 (±8.57) ^b^	10.76 (±6.32) ^ab^

The value in parentheses is the standard deviation. Different letters on the same line show statistical difference (*p* < 0.05). NC: Negative control. AF: Allantoic fluid. UA: Uric Acid. Creat: Creatinine. ALP: Alkaline Phosphatase. GGT: gamma-glutamyl transferase AST: aspartate aminotransferase. ALT: alanine aminotransferase. Ca: calcium. P: phosphorus and CRP: c-reactive protein.

**Table 3 animals-12-01156-t003:** Histopathological analysis of liver from CE infected with *Gammacoronavirus*.

	NC	Virus
Inflammation	0.00 ^a^ (Mi: 0.00; Ma: 0.00)	2.00 ^b^ (Mi: 0.00; Ma: 2.00)
Degeneration	0.00 ^a^ (Mi: 0.00; Ma: 0.00)	0.00 ^a^ (Mi: 0.00; Ma: 0.00)
Necrosis	0.00 ^a^ (Mi: 0.00; Ma: 0.00)	0.00 ^a^ (Mi: 0.00; Ma: 3.00)
Circulatory change	0.00 ^a^ (Mi: 0.00; Ma: 0.00)	1.00 ^b^ (Mi: 0.00; Ma: 3.00)

Different letters on the same line show statistical difference (*p* < 0.05). Mi: Minimum; Ma: Maximum. NC: Negative Control.

**Table 4 animals-12-01156-t004:** Evaluation of CE’s viability, lesions and embryonic mortality when treated with FG and DX at different ages.

		Alive (Normal)	Injured	Dead	Total
Zero EID	NC	6	0	1	6
	FG	5	0	1	6
	DX	0	2	4	6
3 EID	NC	6	0	0	6
	FG	6	0	0	6
	DX	1	5	0	6
7 EID	NC	6	0	0	6
	FG	6	0	0	6
	DX	0	3	3	6
10 EID CAM	NC	5	1	0	6
	FG	6	1	0	7
10 EID SM	CN	6	0	0	6
	DX	3	6	0	9
12 EID	NC	6	0	0	6
	FG	6	0	0	6
	DX	3	3	0	6

Note: eggs broken or killed by the inoculation process were removed from the analysis. CE inoculated at 0, 3, 7, 10 and 12 EID were evaluated at 9, 11, 10, 17 and 19 EID, respectively. EID: Embryonic incubation days. CAM: Chorioallantoic membrane. SM: Shell membrane. NC: Negative control. FG: Filgrastim. DX: Dexamethasone.

**Table 5 animals-12-01156-t005:** Blood cell count in CE treated with FG and DX at 10 EID.

	10 EID			
	NC (CAM)	FG (CAM)	NC (SM)	DX (SM)
Ht (%)	21.64 (±3.38) ^#^	18.43 (±3.60) ^#^	20.33 (±1.86) ^a^	17.13 (±2.35) ^b^
Hg (g/dL) (1/3 ht)	7.20 (±1.13) ^#^	6.14 (±1.20) ^#^	6.78 (±0.62) ^a^	5.71 (±0.78) ^b^
Hg (g/dL) (Cyanmethemoglobin)	8.12 (±0.66) ^#^	7.52 (±1.29) ^#^	7.83 (±0.65) ^a^	7.22 (±1.12) ^b^
Erythrocytes × 10^6^/mm^3^	1.76 (±0.53) ^#^	1.73 (±0.28) ^#^	2.05 (±0.27) ^a^	1.58 (±0.26) ^b^
MCH (pg)	50.34 (±14.07) ^#^	47.35 (±4.95) ^#^	38.35 (±2.04) ^a^	45.58 (±6.69) ^b^
MCV (fL)	107.1 (±13.82) ^#^	135.0 (±43.47) ^#^	99.73 (±8.53) ^a^	109.10 (±13.39) ^a^
MCHC (g/dL)	38.33 (±3.3) ^#^	41.00 (±3.48) ^#^	38.56 (±1.85) ^a^	42.12 (±2.65) ^b^
Thrombocytes × 10^3^/mm^3^	15.8 (±0.97) ^#^	13.3 (±13.1) ^#^	16.3 (±5.1) ^a^	8.75 (±5.75) ^b^
Leucocytes × 10^3^/mm^3^	3.68 (±2.06) ^#^	4.25 (±2.85) ^#^	4.15 (±0.74) ^a^	3.37 (±2.58) ^a^
Monocytes/mm^3^	0.00 (±0.00) ^#^	14.17 (±33.09) ^#^	0.00 (±0.00) ^a^	0.00 (±0.00) ^a^
Lymphocytes/mm^3^	159.01 (±97.72) ^#^	220.80 (±216.80) ^#^	318.5 (±326.3) ^a^	357 (±396.6) ^a^
Heterophiles/mm^3^	3523 (±2020) ^#^	3973 (±2565) ^#^	3104 (±1251) ^a^	2453 (±1695) ^a^
Heterophile/lymphocyte	26.26 (±34.75) ^#^	32.85 (±31.77) ^#^	38.51 (±38.11) ^a^	36.80 (±43.29) ^a^
Eosinophils/mm^3^	5.00 (±15.00) ^#^	25.53 (±69.48) ^#^	0.00 (±0.00) ^a^	1.18 (±6.03) ^a^
Basophils/mm^3^	0.00 (±0.00) ^#^	0.00 (±0.00) ^#^	0.00 (±0.00) ^a^	0.00 (±0.00) ^a^

Statistical comparisons are between FG CAM and NC CAM (test t) or DX SM and NC SM (test t). Different symbols on the same line indicate a statistical difference between FG and NC inoculated via CAM. Different lowercase letters on the same line indicate a statistical difference between DX and NC at inoculated via SM (*p* < 0.05). MCH, eosinophils, monocytes; via CAM: non-parametric test. Ht: Hematocrit. Hg: Hemoglobin. MCH: corpuscular volume. MCH: mean corpuscular hemoglobin. MCHC: mean corpuscular hemoglobin concentration.

**Table 6 animals-12-01156-t006:** Blood cell count in CE treated with FG and DX at 12 EID.

	12 EID		
	CN (SM)	FG (SM)	DX (SM)
Ht (%)	27.80 (±3.56) ^A^	28.67 (±3.50) ^A^	23.40 (±3.71) ^A^
Hg (g/dL) (1/3 ht)	8.93 (±1.03) ^A^	9.55 (±1.17) ^A^	9.26 (±1.48) ^A^
Hg (g/dL) (Cyanmethemoglobin)	10.61 (±3.04) ^A^	10.55 (±1.48) ^A^	7.90 (±1.13) ^A^
Erythrocytes × 10^6^/mm^3^	2.59 (±0.48) ^A^	2.66 (±0.38) ^A^	1.90 (±0.26) ^B^
MCH (pg)	41.13 (±9.12) ^A^	39.65 (±2.12) ^A^	41.09 (±1.39) ^A^
MCV (fL)	102.8 (±25.82) ^A^	108.4 (±10.98) ^A^	128.6 (±10.29) ^A^
MCHC (g/dL)	37.75 (±7.09) ^A^	36.92 (±4.40) ^A^	33.28 (±3.18) ^A^
Thrombocytes × 10^3^/mm^3^	5.30 (±1.35) ^A^	5.66 (±1.08) ^A^	14.5 (±7.32) ^B^
Leucocytes × 10^3^/mm^3^	4.00 (±1.05) ^A^	4.96 (±1.78) ^A^	16.3 (±8.30) ^B^
Monocytes/mm^3^	45.83 (±40.79) ^AB^	61.67 (±34.86) ^A^	40.00 (±132.70) ^B^
Lymphocytes/mm^3^	457.60 (±114.80) ^A^	516.70 (±139.00) ^A^	271.90 (±139.80) ^B^
Heterophiles/mm^3^	3498 (±940.50) ^A^	4132 (±1549) ^A^	16,900 (±8200) ^B^
Heterophile/lymphocyte	8.00 (±1.55) ^A^	7.00 (±3.30) ^A^	65.77 (±29.40) ^B^
Eosinophils/mm^3^	0.00 (±0.00) ^A^	0.00 (±0.00) ^A^	0.00 (±0.00) ^A^
Basophils/mm^3^	0.00 (±0.00) ^A^	0.00 (±0.00) ^A^	0.00 (±0.00) ^A^

Statistical comparisons are between NC, FG, DX inoculated via SM (ANOVA). Different uppercase letters on the same line indicate a statistical difference (*p* < 0.05). Leucocytes, lymphocytes and monocytes FG via CAM: non-parametric test. Ht: Hematocrit. Hg: Hemoglobin. MCH: corpuscular volume. MCH: mean corpuscular hemoglobin. MCHC: mean corpuscular hemoglobin concentration.

**Table 7 animals-12-01156-t007:** Quantification of metabolites, minerals and enzymes in serum and AF in CE treated with FG and DX at 10EID inoculated via CAM.

	10 EID			
	CAM			
	NC (Serum)	FG (Serum)	NC (AF)	FG (AF)
UA (mg/dL)	11.00 (±7.60) ^#^	16.49 (±14.55) ^#^	97.57 (±31.16) *	87.88 (±51.40) *
Creat (mg/dL)	1.31 (±0.68) ^#^	0.80 (±0.63) ^#^	2.49 (±0.85) *	3.19 (±0.77) *
APL (U/L)	1836 (±815.30) ^#^	2380 (±298.40) ^#^	47.05 (±32.58) *	61.26 (±53.26) *
GGT (U/L)	63.17 (±85.13) ^#^	67.00 (±81.56) ^#^	14.53 (±5.50) ^#^	12.49 (±6.98) ^#^
AST (U/L)	48.12 (±29.53) ^#^	49.18 (±15.01) ^#^	15.14 (±11.55) *	9.33 (±6.02) *
ALT (U/L)	54.02 (±51.67) ^#^*	158.00 (±150.10) ^#^	8.28 (±4.07) *	11.71 (±8.51) *
Ca (mg/dL)	18.80 (±17.03) ^#^	21.80 (±21.75) ^#^	12.50 (±7.05) ^#^	8.34 (±4.62) ^#^
P (mg/dL)	7.48 (±4.52) ^#^	4.52 (±2.58) ^#^	15.37 (±9.04) ^#^*	22.13 (±9.63) *

Statistical comparisons are between FG CAM and NC CAM (*t* test) serum and AF inoculated at 10 EID. Different symbols on the same line indicate a statistical difference between FG and NC inoculated at 10 EID via CAM. Creat in FG group inoculated at 10 EID: non-parametric test. UA: Uric Acid. Creat: Creatinine. ALP: Alkaline Phosphatase. GGT: gamma-glutamyl transferase AST: aspartate aminotransferase. ALT: alanine aminotransferase. Ca: calcium. P: phosphorus and CRP: c-reactive protein.

**Table 8 animals-12-01156-t008:** Quantification of metabolites, minerals and enzymes in serum and AF in CE treated with FG and DX at 10EID inoculated via SM.

10 EID				
	SM			
	NC (Serum)	DX (Serum)	NC (AF)	DX (AF)
UA (mg/dL)	13.86 (±10.37) ^ab^	12.04 (±9.84) ^a^	60.00 (±38.63) ^ab^	70.30 (±45.93) ^b^
Creat (mg/dL)	0.53 (±0.35) ^a^	1.12 (±1.03) ^a^	3.57 (±0.53) ^b^	2.62 (±1.29) ^b^
APL (U/L)	2125 (±1365) ^a^	4233 (±5830) ^a^	45.94 (±39.65) ^b^	77.81 (±54.46) ^b^
GGT (U/L)	81.20 (±94.30) ^a^	49.41 (±82.50) ^a^	12.72 (±11.31) ^a^	19.51 (±9.54) ^a^
AST (U/L)	37.55 (±20.84) ^a^	99.89 (±68.10) ^b^	39.87 (±55.81) ^a^	13.27 (±4.25) ^a^
ALT (U/L)	142.6 (±161.3) ^a^	142.8 (±138.7) ^a^	8.00 (±1.58) ^a^	13.00 (±8.69) ^a^
Ca (mg/dL)	9.15 (±10.65) ^a^	30.53 (±30.50) ^a^	6.50 (±2.80) ^a^	15.91 (±13.46) ^a^
P (mg/dL)	4.68 (±3.33) ^a^	5.80 (±4.21) ^a^	23.65 (±6.32) ^b^	26.90 (±11.91) ^b^

Statistical comparisons are between DX SM and NC SM (*t* test) inoculated at 10 EID (*t* test) serum and AF. Different lowercase letters on the same line indicate a statistical difference between DX and NC inoculated at 10 EID via SM. GGT in DX group inoculated at 10 EID via SM: non-parametric test. GGT and ALP in DX group inoculated at 10 EID via SM; Creat in FG group inoculated at 10 EID: non-parametric test. UA: Uric Acid. Creat: Creatinine. ALP: Alkaline Phosphatase. GGT: gamma-glutamyl transferase AST: aspartate aminotransferase. ALT: alanine aminotransferase. Ca: calcium. P: phosphorus and CRP: c-reactive protein.

**Table 9 animals-12-01156-t009:** Quantification of metabolites, minerals and enzymes in serum and AF in CE treated with FG and DX at 12 EID inoculated via SM.

	12 EID		
	SM		
	CN (Serum)	FG (Serum)	DX (Serum)
UA (mg/dL)	8.28 (±1.88) ^A^	12.53 (±10.17) ^A^	9.50 (±10.82) ^A^
Creat (mg/dL)	1.37 (±1.04) ^A^	0.70 (±0.53) ^A^	1.39 (±0.53) ^A^
APL (U/L)	2194 (±322.90) ^A^	2439 (±437.70) ^A^	1928 (±356.40) ^A^
GGT (U/L)	47.25 (±40.85) ^A^	59.84 (±64.05) ^A^	69.08 (±47.69) ^A^
AST (U/L)	123.1 (±81.82) ^A^	108.2 (±19.21) ^A^	556.1 (±27.61) ^A^
ALT (U/L)	56.68 (±59.33) ^A^	130.5 (±226.9) ^A^	73.70 (±25.15) ^A^
Ca (mg/dL)	17.98 (±9.35) ^A^	17.70 (±12.28) ^A^	13.63 (±10.32) ^A^
P (mg/dL)	5.22 (±3.89) ^A^	4.76 (±2.85) ^A^	9.52 (±3.28) ^A^

Statistical comparisons are between NC, FG, DX inoculated at 12 EID via SM (ANOVA) in serum because at 19 EID (age at colect) there are not AF. Different uppercase letters on the same line indicate a statistical difference in CE inoculated at 12 EID. UA, ALT, group inoculated at 12 EID: non-parametric test. UA: Uric Acid. Creat: Creatinine. ALP: Alkaline Phosphatase. GGT: gamma-glutamyl transferase AST: aspartate aminotransferase. ALT: alanine aminotransferase. Ca: calcium. P: phosphorus and CRP: c-reactive protein.

**Table 10 animals-12-01156-t010:** Histopathological analysis of liver from embryos inoculated with FG and DX at 10 and 12 EID.

	10 EID				12 EID		
	NC (CAM)	FG (CAM)	NC (SM)	DX (SM)	NC	FG	DX
Inflammation	0 (Mi: 0; Ma: 0) ^#^	1 (Mi: 0; Ma: 2) *	0 (Mi: 0; Ma: 0) ^a^	0 (Mi: 0; Ma: 2) ^b^	0 (Mi: 0; Ma: 0) ^A^	0 (Mi: 0; Ma: 1) ^AB^	1 (Mi: 0; Ma: 2) ^B^
Degeneration	0 (Mi: 0; Ma: 0) ^#^	0 (Mi: 0; Ma: 0) ^#^	0 (Mi: 0; Ma: 0) ^a^	0 (Mi: 0; Ma: 3) ^a^	0 (Mi: 0; Ma: 0) ^A^	0 (Mi: 1; Ma: 1) ^AB^	1 (Mi: 1; Ma: 3) ^B^
Necrosis	0 (Mi: 0; Ma: 0) ^#^	0 (Mi: 0; Ma: 0) ^#^	0 (Mi: 0; Ma: 0) ^a^	0 (Mi: 0; Ma: 2) ^a^	0 (Mi: 0; Ma: 0) ^A^	0 (Mi: 0; Ma: 0) ^A^	0 (Mi: 0; Ma: 0) ^A^
Circulatory change	0 (Mi: 0; Ma: 0) ^#^	0 (Mi: 0; Ma: 1) ^#^	0 (Mi: 0; Ma: 0) ^a^	1 (Mi: 0; Ma: 3) ^b^	0 (Mi: 0; Ma: 0) ^A^	0 (Mi: 0; Ma: 1) ^A^	0 (Mi: 0; Ma: 0) ^A^

Different symbols on the same line indicate a statistical difference between FG and NC inoculated via CAM. Different lowercase letters on the same line indicate a statistical difference between DX and NC at inoculated via SM. Different uppercase letters on the same line indicate a statistical difference in CE inoculated at 12 EID (*p* < 0.05). Mi: Minimum, Ma: Maximum. NC: Negative control. CAM: Chorioallantoic membrane. FG: Filgrastim. SM: Shell Membrane. DX: Dexamethasone. EID: Embryonated incubation days.

## Data Availability

Data are contained within the article or Appendix A. Full data will be made available if the publisher requests it.

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
