# Peer review of "Physiological Changes in Chicken Embryos Inoculated with Drugs and Viruses Highlight the Need for More Standardization of this Animal Model"

_animals, 2022, doi:10.3390/ani12091156_

Round 1

Reviewer 1 Report

This is a very well-written and well-described manuscript demonstrating the need for standardization of current experimental protocols when using chicken embryos as a model. The introduction covers the current knowledge well. The methods section is appropriately subdivided into smaller sections, allowing replication of each section by itself if required by other researchers. The results are well-presented, with the longer than normal length of the section simply a reflection of the large amount of data generated by the treatments studied. The discussion summarizes the study well, and highlights important steps moving forward.

Reviewer 2 Report

  1. Abbreviations in the abstract and other sections should be revised.
  2. Writing mistakes have been identified in many sections.
  3. There is spelling error in the paragraphs and references.
  4. In the references section there are rule and spelling mistakes.

For instance; chicken embryo (CE) or chicken embryos (CE) which one?(singular, Plural?)

                        didn’t or did not (page 1)

                        specific to can be used instead of peculiar to. (p.1)

                         increased can be uesd instead of augmented (p.22)

                        gallus gallus embryo can be used instead of chicken embryo (p.1)

                        depended on can be used instead of depending on (p.1)

                        dependent on can be used instead of dependent on (p23).

                        ºC or oC (p.2), −80 ºC, 4 ºC, 37 ºC  (p.5)

                              the allantoid fluid or the allantoic fluid? (p.2)

                        two-hour or two hour (p.2)

                        0.5mL or 0.5 ml? (p.2), 5 mMTris or 5 mM Tris (p.5)

2-4 mg/Kg or 2-4 mg/kg; 5 mg/Kg or 5 mg/kg; ~4 μg/Kg; ~7.5 μg/Kg (p.3)

CETest or CE Test (p.3)

Haematocrit or hematocrit, haemoglobin or hemoglobin (p.4), sulphydryl or sulfhydryl (p.5,8,15, 22)

Not were, it’s where (p.20).

Calcium-binding proteins was; not was, it’s plural that’ s were (p.20).

ın granulocytes was; not was, it’s plural that’s were (p21)

macro and microscopic damage was; not was, it’s plural that’s were (p.23)

Person’s correlation or Pearson correlation? (p.6)

non-parametric test.. (two dots??) (p.15)

30,000/mm3; < 4,000/mm3.  mm3 or mm3 (p.19)

Ca2 or Ca+2 (p.20)

anaemic or anemic (p.20)

function27and , it’s wrong. Delete 27 number. (p.21)

bursa of fabricius or only bursa fabricius

not cannot, can not (p.22)

  1. The abbreviated word at the beginning cannot be rewritten in its clear form.

                        Chicken embryo (CE) or only CE (p.2)

                        Chorioallantoic membran (CAM) (p.3)

                        Haemoglobin or Hb (p.4)

                        Haematocrit or Ht(p.4)

                        not Aspartate aminotransferase (AST), only AST (p.22)

                        not Alkaline phosphatase (ALP), only ALP (p.22)

not filgrastim, only FG (p.22)

  1. Why are titles written in italic form?

                        2.1. Challenge of CE with Gammacoronavirus? (p.2)

  1. Do not use dots in titles

3.6. The amount of metabolites, enzymes and minerals in serum and allantois are not always 406 identical. (p.13)

  1. Sentence errors should be checked

The response of the white blood cell count during an infection can vary. Or During an infection the response of the white blood cell count can vary. (p.20)

In the initial moment of disease or In the initial of disease (p.20).

by the first three days or within first three days

  1. The reference section should be reviewed.
  2. Lesions on histopathological images should be shown more clearly.

11. Abstract is not concise. The purpose of the study should fully reflect.It should be a little shorter and simpler. Why is there Simple Summary?

Reviewer 3 Report

This study aimed to verify if the results of viability, weight, pathological and histopathological changes, blood count, the dosage of metabolites and/or enzymes, and oxidative stress in the chicken embryos of different ages are peculiar to the model, proposing and discussing the need for standardization for embryos, using a virus (Gammacoronavirus) and two drugs (dexamethasone and filgrastim). This study is within the scope of Animals. And it is a valuable addition to reinforce the importance of standardization and correct use of the chicken embryos model. This manuscript can be accepted but need further modifications. Some of my specific comments are listed below:

  1. Abstract: The current Results section is too general and should be presented in more detail.

  1. Why did the authors choose these two drugs (dexamethasone and filgrastim)? In the introduction, the authors should briefly describe the two drugs.

  1. Lines 85-87: How did the authors determine such an inoculation dose? Are there any references? Or did you do a pre-experiment?

Reviewer 4 Report

Scientific research has gone beyond animal models

Author Response

This manuscript is a resubmission of an earlier submission. The following is a list of the peer review reports and author responses from that submission.